*Method*

# Rescuing error control in crosslinking mass spectrometry

Lutz Fischer [ID][1] & Juri Rappsilber [ID][1,2,3]✉

## Abstract

Crosslinking mass spectrometry is a powerful tool to study protein-protein interactions under native or near-native conditions in complex mixtures. Through novel search controls, we show how biasing results towards likely correct proteins can subtly undermine error estimation of crosslinks, with significant consequences. Without adjustments to address this issue, we have misidentified an average of 260 interspecies protein-protein interactions across 16 analyses in which we synthetically mixed data of different species, misleadingly suggesting profound biological connections that do not exist. We also demonstrate how data analysis procedures can be tested and refined to restore the integrity of the decoy-false positive relationship, a crucial element for reliably identifying protein-protein interactions.

**Keywords** Crosslinking Mass Spectrometry; Proteomics; Error Estimation; Data Analysis; Data Reliability
**Subject Category** Proteomics

## Introduction

Crosslinking mass spectrometry (MS) has emerged as a powerful approach for studying protein-protein interactions in native or near-native conditions (O'Reilly and Rappsilber, 2018; Piersimoni et al, 2022). This technique involves introducing a crosslinker into a protein sample to covalently connect interacting proteins, followed by digesting the sample and identifying the linked peptide pairs through mass spectrometry. The challenge in identifying these crosslinked peptides arises from several factors, particularly the vast database search space required. Additionally, crosslinked peptide pairs between different protein sequences (referred to as protein heteromeric links in the HUPO PSI controlled vocabulary) are less abundant than crosslinks between peptides within one protein sequence (self crosslinks). These heteromeric links are subject to higher rates of random matching due to their scarcity and the increased number of theoretical combinations in the search space, resulting in lower scores and

increased noise levels—the proportion of random matches in these groups. Therefore, accurately distinguishing true heteromeric crosslinks from false positives is challenging, which limits the sensitivity of crosslinking MS.

To enhance the identification of protein heteromeric crosslinks, it may be tempting to utilise information beyond individual crosslink-spectrum matches (CSMs). Given that protein abundance will impact the likelihood of a protein being observed, it seems perfectly reasonable to accordingly restrict the data analysis. One example of such a strategy, referred to as mi-filter (Chen et al, 2022) (Fig. 1), considers only those proteins that also show self-crosslinking or linear peptides modified by a crosslinker (monolinks) to reduce the false positives among the heteromeric matches. This is based on the observation that monolinks and self crosslinks are more abundant and, hence, more detectable than heteromeric crosslinks. Other tools like ECL-PF (Zhou et al, 2023), CRIMP 2.0 (Crowder et al, 2023), XLinkProphet (Keller et al, 2019) and likely others similarly make use of information about individual proteins during search or rescoring of matches. However, leveraging these observations can be tricky, and improper application could undermine the integrity of the error models used in decoy-based false discovery rate (FDR) assessments.

Typically, crosslinking mass spectrometry uses decoy matches—known false positives—to estimate the prevalence of unknown false positives among the target matches. This approach allows for filtering results to a predefined confidence level, reflected as a false discovery rate (FDR) (Maiolica et al, 2007; Walzthoeni et al, 2012; Yang et al, 2012; Fischer and Rappsilber, 2017, 2018; Lenz et al, 2021). Decoy matches typically involve searching either reversed target proteins (Maiolica et al, 2007) or randomly generated proteins (Kaake et al, 2014), where each decoy protein serves as an equivalent to a target protein. The quantity and score distribution of decoy matches provides a model for the potential false positives among the target matches.

However, we demonstrate that approaches that filter search results based on additional considerations can disadvantage decoys relative to targets, thereby impairing the decoys' ability to model false positive targets. Consequently, these remaining decoys cannot be reliably used to gauge the confidence of results. Moreover, we outline how to increase the detection of protein-protein interactions at a given confidence level, without underestimating the incidence of false positives when adding protein level information.

[1]Technische Universität Berlin, Chair of Bioanalytics, 10623 Berlin, Germany. [2]Wellcome Centre for Cell Biology, University of Edinburgh, Edinburgh EH9 3BF, UK. [3]Si-M/"Der Simulierte Mensch", a Science Framework of Technische Universität Berlin and Charité - Universitätsmedizin Berlin, Berlin, Germany. ✉E-mail: juri.rappsilber@tu-berlin.de

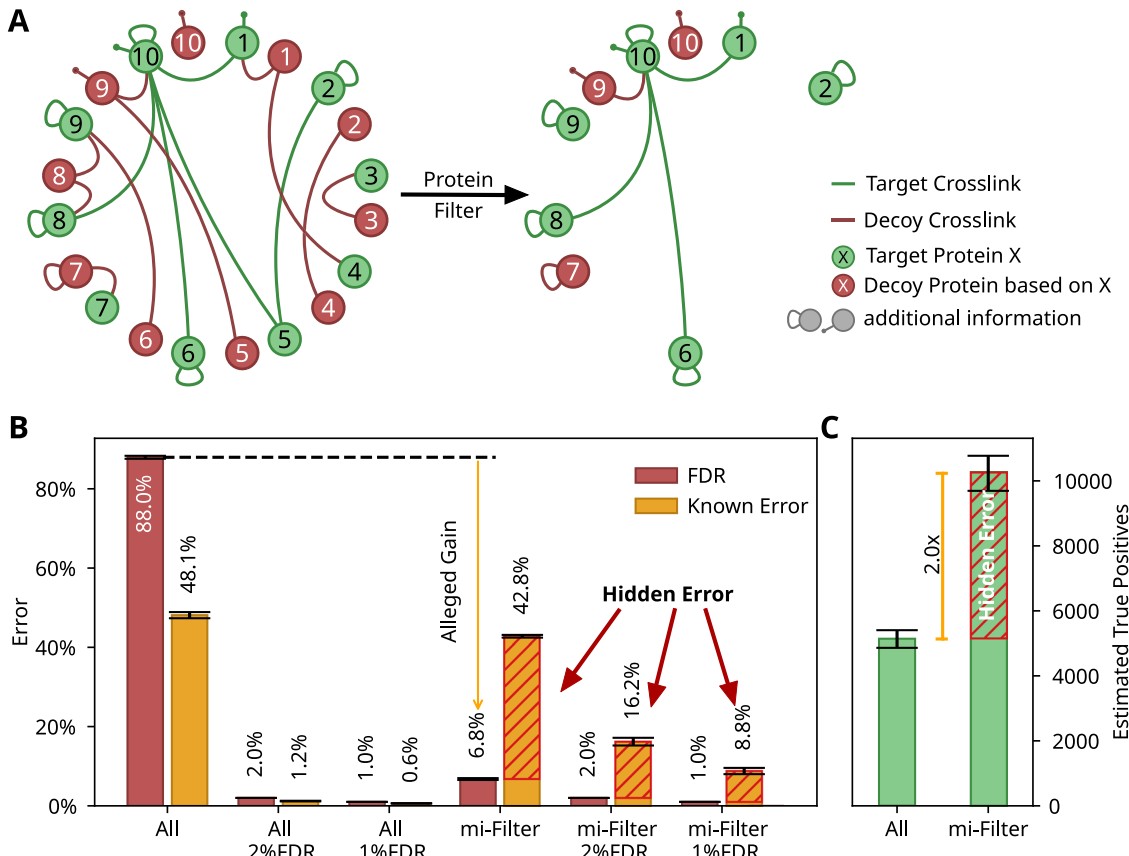

**Figure 1. A simple protein-based filter leads to hidden error.**

(**A**) Schema of a protein-based filter that considers information on individual proteins. Only proteins that pass the filter are considered further. In the case of the mi-filter, only proteins that are observed with monolinks or self crosslinks are kept and considered as observable in heteromeric protein-protein interactions. (**B**) Effect of the mi-filter on false positives: decoy-based FDR estimate (green) and known error (orange). The known error comprises the matches to *E. coli* spectra that involve *M. pneumoniae* proteins and matches to *M. pneumoniae* spectra that involve *E. coli* proteins. A minimum score cut-off of 5 was applied, resulting in 42,399[ ± 1485] target matches and 20336[ ± 669] known false positives (All), which changed after applying the mi-filter to 10,966[ ± 568] target matches and 4685[ ± 236] known false positives. The mi-filter results in a known error of 16.2% (267[ ± 27] of 1625[ ± 112] target matches) and 9% (104[ ± 12] of 1160[ ± 82] target matches) at 2 and 1% FDR cut-offs. The shaded areas indicate the fraction of known error not modelled by decoys (i.e. hidden error). (**C**) Effect of the mi-filter on the total estimated true positive: Decoy-based estimate of assumed true positives among results for either just applying a minimum score cut-off (left) or after applying the score cut-off and mi-filter (right). The shaded area indicates the number of impossible true positives. Data plotted are the average for 16 pairwise combinations of each of four SCX fractions of *M. pneumoniae* with each of four SCX fractions of *E. coli* fractions. Error bars indicate the standard error.

# Methods

### Reagents and tools table

| Software | |
| --- | --- |
| xiSEARCH 1.7.6.7 | https://www.rappsilberlab.org/software/xisearch/ |
| xiFDR 2.0 | https://www.rappsilberlab.org/software/xifdr/ |
| msConvert (ProteoWizard 3.0) | https://proteowizard.sourceforge.io/ |

## Dataset origins

The datasets employed in this study were obtained from ProteomeXchange (Deutsch et al, 2023). The *Mycoplasma pneumoniae* dataset, specifically raw files from strong cation exchange (SCX) fractions

11 to 14, was sourced from Pride ID PXD017711 (O'Reilly et al, 2020; Data ref: O'Reilly and Rappsilber, 2020). The *E. coli* dataset came from JPOST ID JPST000845 (Lenz et al, 2021; Data ref: Sinn and Rappsilber, 2021), comprising DSSO raw files for SCX fractions 18, 20, 22, and 24. Additionally, the 26S proteasome dataset included the trypsin-only raw files from Pride ID PXD008550 (Mendes et al, 2019; Data ref: Mendes and Rappsilber, 2019).

## Data processing

The raw files were converted to mgf-files with msConvert from ProteoWizard (version 3.0) (Chambers et al, 2012) with peak picking enabled. Crosslink search was done with xiSEARCH (version 1.7.6.4). Search parameters used were: crosslinker DSSO for *M. pneumoniae* and *E. coli* dataset and BS3 for the 26S proteasome dataset with specificity for lysine, serine, threonine, tyrosine and protein n-terminal with a penalty value for serine, threonine, and tyrosine of 0.2, fixed

modification of carbamidomethylation of Cysteine, variable modification of oxidation on methionine and hydrolysed and amidated crosslinker modifications on K, S, T, Y and protein N-termini as linear only modifications. Non-covalent interactions were considered as part of the search as well but ignored during data analysis.

The datasets from *E. coli* and *M. pneumoniae* were analysed using a combined database comprising all *E. coli* proteins (UniProt proteome UP000000625 as of January 19, 2023) and all *M. pneumoniae* proteins (UniProt proteome UP000000808 as of January 16, 2023). This approach of searching against a unified database serves as a decoy-independent method for generating a set of known false positives, providing a benchmark to evaluate if decoy matches accurately mirror false positive target matches. Specifically, any detected match between *E. coli* and *M. pneumoniae* proteins, or an *M. pneumoniae* protein matched to an *E. coli* spectrum and vice versa, is inherently incorrect. This methodology is akin to a traditional entrapment strategy, where a dataset is searched against both target and non-present (entrapment) protein sequences. The advantage of the pairwise entrapment model is that all proteins act simultaneously as targets for some spectra and as known false proteins for others. For data analysis, spectra from each *E. coli* SCX fraction ($n = 4$) were paired with those of each *M. pneumoniae* SCX fraction ($n = 4$), and the average number of matches of all possible combinations ($n = 16$) was calculated and plotted.

The 26S proteasome dataset was initially processed using MaxQuant (Tyanova et al, 2016) version 1.6.17, targeting the complete Saccharomyces cerevisiae proteome (UniProt proteome UP000002311 as of February 6, 2023). This analysis identified 1073 non-contaminant target protein groups, from which the first protein of each group was selected as a representative. These proteins were then ranked by their iBAQ values and divided into three sets. Proteins not identified in the original FASTA file were grouped into ten subsets.

For the crosslinking MS data analysis, xiSEARCH (version 1.7.6.4) was initially used to search against a progressively increasing number of present proteins, sorted by abundance. To evaluate the impact of the filter with non-present proteins, the search included all previously identified proteins plus incremental additions of non-identified protein sets. The results were then filtered using xiFDR 2.2, with experiments conducted both with and without boosting on residue pairs, and with the ec-filter enabled and disabled. Data analysis was performed by individually searching and filtering each fraction.

To enable a valid FDR calculation, the datasets were filtered to only accept the highest scoring match for any given combination of peptide pairs, precursor charge state modifications and linkage-site—termed unique CSM in xiFDR.

# Results

## Classes of false positives

In the context of protein-based filters in crosslinking MS, we can describe two different types of false positive matches within the realm of target matches:

**False Positive Group 1:** This category encompasses random matches that involve at least one protein that is not observable as part of a genuine crosslink. This situation arises either because the protein is absent from our sample or due to practical factors like low protein abundance, rendering it undetectable as part of a

crosslink. If such a protein is nevertheless identified by being matched to a spectrum, it constitutes a false positive. Of course, one does not know which specific matches this applies to. However, one knows that this error occurs.

**False Positive Group 2:** This category encompasses random matches between protein pairs where both proteins are observable as part of genuine crosslinks. This means that these proteins are detectable as interacting with each other or with one or more other proteins. Being part of a genuine crosslink does not mean that a protein can not also be matched in a false positive peptide-spectrum match. In other words, a random match may still occur involving a protein with multiple correctly identified crosslinks. For example, if two protein pairs, AB and XY, truly existed independently of each other in a sample, one could still identify false matches between them, i.e., AX, AY, BX, BY, AA, BB, XX, and YY.

In the absence of any rescoring or filtering, decoys model both of these false positive groups. However, this conventional approach becomes inadequate when a filter is introduced that distinguishes between these two groups, as it necessitates a more nuanced modelling strategy.

## Theoretical weakness of post-search protein discrimination

One way of post-search protein discrimination is by limiting protein-protein crosslinks to only those proteins that were observed also with self crosslinks or monolinks. This is done by the mi-filter (Fig. 1A). Related heuristics have been employed before, albeit when constructing the search database. These include restricting the search to those proteins that are in the sample, i.e. can be identified by any peptide (Maiolica et al, 2007; Götze et al, 2019) or those proteins that are identified with a certain abundance (Mendes et al, 2019; Lenz et al, 2021). Asking for self crosslinks or monolinks relates to the abundance criterion, as these peptides tend to have higher abundance than protein heteromeric crosslinks but lower abundance than linear unmodified peptides. One also adds the observation that peptides of those proteins actually reacted with the crosslinker, although, it is unclear if this is important. In this way, the mi-filter excludes proteins that are likely not observable as being crosslinked with other proteins. This reduces noise matches. Note that the mi-filter also reduces correct matches by biasing against proteins with few identifiable peptides, for example, small proteins (Lenz et al, 2021).

However, because the mi-filter is applied as a filter after the search and not before, it requires some consideration of how target and decoy proteins are affected. The passing decoys now model proteins that pass the filter falsely, i.e. are not observable as self-link or monolink. In line with the initial hypothesis, these can be assumed to be not observable as part of a heteromeric link either. As a result, after filtering the protein heteromeric matches, any decoys can now only represent false positive matches from the false positive group 1. However, matches from false positive group 2 remain present. Consequently, any FDR calculation relying on decoys will underestimate the total error.

## Post-search protein discrimination can lead to severely underestimating the error

To assess if post-search protein discrimination, as done by the mi-filter, leads to underestimation of error requires a

decoy-independent test of error. An entrapment search is used frequently in proteomics for similar evaluations, and has already been used in the context of crosslinking (Lenz et al, 2021). This method involves adding a set of sequences to the database that are known not to be present in the sample being analysed—these are the "entrapment" sequences. When the mass spectrometry data is searched against this augmented database, any identifications matching the entrapment sequences can be confidently classified as false positives because these sequences do not exist in the experimental sample. Unfortunately, a simple entrapment search only provides ground truth for the absence of proteins, and, in effect, can only test if false positive group 1 (crosslinks involving non-present proteins) is modelled.

We therefore develop here the pairwise entrapment search, by constructing a test case of two sets of proteins that are crosslinked and measured only within each set but searched together, permitting "identifications" of crosslinks also between the sets of proteins. This allows us to reveal false protein pairings among actually present and crosslinkable proteins (false positive group 2 errors or types AX, AY, BX, and BY in the example above). For this, we took the data of two separate large-scale crosslink investigations, from *E. coli* (Lenz et al, 2021) and *M. pneumoniae* (O'Reilly et al, 2020), and searched against a combined database of *E. coli* and *M. pneumoniae* proteins. In this way, the *M. pneumoniae* proteins become the entrapment database for the *E. coli* data and vice versa. Importantly, both species contain observable proteins that, at the same time, will be visibly false positive when they are matched to spectra of the other species or in a pair together with a protein of the other species. Our pairwise entrapment setup establishes a baseline for identifying proteins in cases where they should not crosslink or correspond to a given spectrum. This approach more accurately mirrors the complexity and dynamic range of real biological experiments compared to synthetic models of peptides or proteins. Our approach is reminiscent of studying a eukaryotic cell, where proteins separate into subsets (such as compartments). We have, however, then only moderate confidence in the composition of these protein subsets. In contrast, our method provides definitive information on distinct protein groups.

Examining all matches that meet a minimum score threshold (xiSEARCH score >= 5), the decoy-based estimation of false positives surpasses the count of observed impossible matches (Fig. 1B). This discrepancy is anticipated since legitimate matches (e.g. *E. coli* protein pairs matched to *E. coli* spectra and *M. pneumoniae* protein pairs matched to *M. pneumoniae* spectra) inevitably include incorrect results (false positives). However, after applying the mi-filter, the number of remaining decoys drops dramatically, leading to a significantly reduced estimate of false positives. Nevertheless, the frequency of impossible matches—those that cross dataset boundaries—is substantially higher than expected if the false discovery rate (FDR) estimates were accurate. Specifically, there are more than five times as many impossible matches as there are decoy-estimated false positives. Consequently, the actual FDR for mi-filtered results must exceed 33%, even though the decoys suggest an apparent FDR of only 6.2% at the crosslink-spectrum match level.

This indicates that much of the perceived benefit from the mi-filter may merely mask the true error rate (illustrated in Fig. 1B with a red hatched area). Implementing an FDR-based cutoff such as 1 or 2% for decoys would still result in 8.6 or 15.9% impossible

matches, respectively, indicating a significantly greater error in the reported data than anticipated. As a result, we could erroneously report extensive protein-protein interactions between *E. coli* and *M. pneumoniae*—in our 16 test analyses, an average of 260 interspecies PPIs based on 2% CSM-FDR or 67 at 2% PPI-FDR—suggesting profound biological connections that do not actually exist. These conclusions would stem from a critical error in data analysis.

## A universal test for error estimates being affected by filters

A more comprehensive strategy to evaluate whether a filter disrupts the estimation of false positives would involve changing the viewpoint from false positives to true positives. The number of true positives can be estimated by subtracting the estimated number of false positives from all target-target matches:

$$eTP_{TT} = TT - eFP_{TT} \qquad (1)$$

With $eTP_{TT}$ being the number of estimated true positives, $eFP_{TT}$ being the estimated false positive matches and TT being the total number of matches in which the amount of false and true positives are to be estimated.

As we here use decoys to model false positive crosslink matches, $eFP$ turns into (Walzthoeni et al, 2012; Fischer and Rappsilber, 2017)

$$eFP_{TT} = TD - DD \qquad (2)$$

With $eFP_{TT}$ representing the estimated number of false positives among the target-target matches, TD the number of matches involving one target and one decoy part, and DD the number of matches that involve only decoys. The possibly surprising subtraction of DD from TD in this formula is the result of search space considerations (Fischer and Rappsilber, 2017).

Inserting (2) into our initial formula (1) results in:

$$eTP_{TT} = TT - (TD - DD) \qquad (3)$$

With TT being the number of matches that fall into the target database.

Assuming the method used to estimate the number of false positives (in this case, based on decoys) is accurate, our formula reveals a theoretical maximum on the number of true positives that can be identified. Therefore, the count of estimated true positives after applying any filter or other approach should not exceed that found in the unfiltered dataset. Typically, most correct filtering approaches reduce the total number of true positives. However, after applying the mi-filter, the estimated number of true positive crosslink-spectrum matches (CSMs) is three times higher than that in the unfiltered dataset (Fig. 1C). This significant discrepancy suggests a serious overestimation of true positives and an accompanying underestimation of false positives.

It is important to recognise that this test primarily identifies potential issues; passing this test does not guarantee the correctness of the filtering method. Furthermore, the test's effectiveness hinges on directly comparing the input data with the data that undergoes the specific filter. Introducing further processing steps like FDR

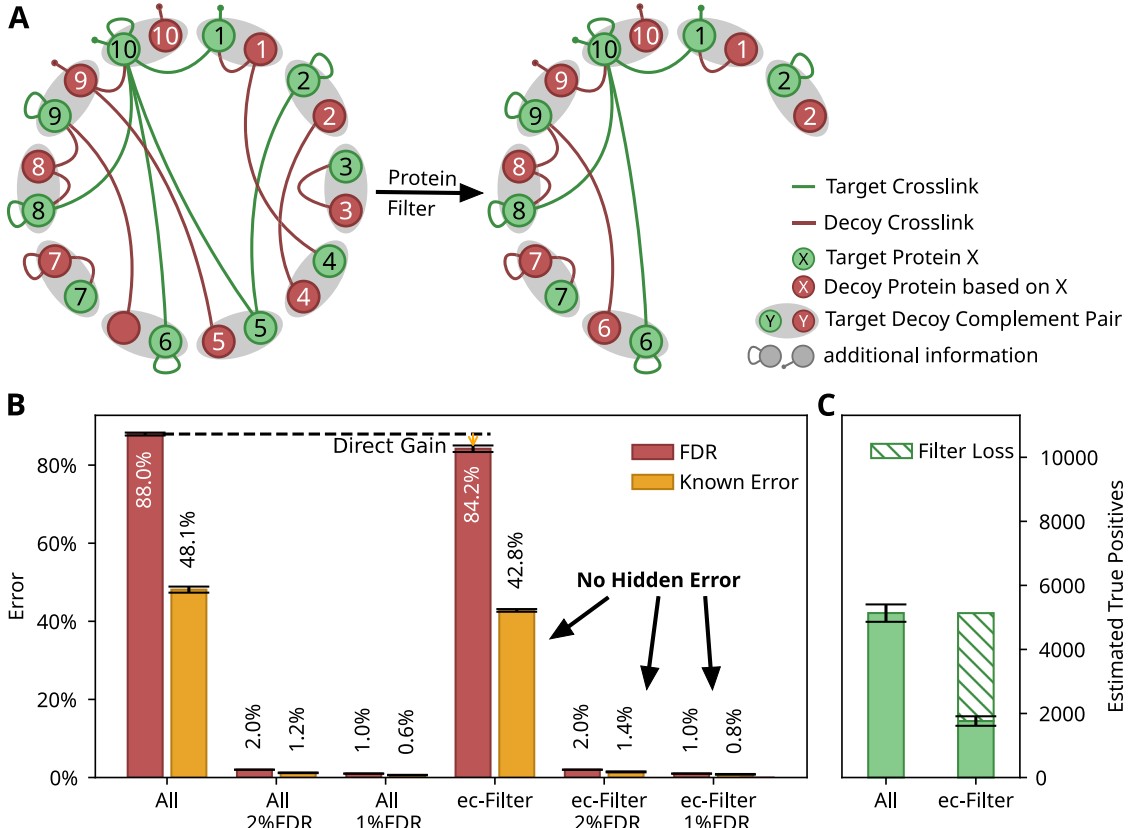

**Figure 2. Protein-based filter with target-decoy pairing.**

(A) Schema of the ec-filter as a protein-based filter with target-decoy pairing: Proteins are treated as target-decoy pairs and these are filtered by the ec-filter considering self crosslinks or monolinks in either partner. (B) Effect of the ec-filter on false positives: decoy-based FDR estimate (green) and known error (orange). The known error comprises the matches to *E. coli* spectra that involve *M. pneumoniae* proteins and matches to *M. pneumoniae* spectra that involve *E. coli* proteins. A minimum score cut-off of 5 was applied, resulting in 42,399[ ± 1485] target matches and 20,336[ ± 669] known false positives (All), which changed after applying the ec-filter to 10,966[ ± 568] target matches and 4685[ ± 236] known false positives. The ec-filter results in no hidden error by returning 1.4% (9.7[ ± 1] of 672[ ± 47]) and 0.8% (4.8[ ± 0.8] of 546[ ± 47]) known error, respectively, when aiming for 2 and 1% FDR based on decoys. (C) Decoy-based estimate of assumed true positives among results for either just applying a minimum score cut-off (left) or after applying the score cut-off and ec-filter (right). The shaded area indicates the number of impossible true positives. Data plotted are the average for 16 pairwise combinations of each of four SCX fractions of *M. pneumoniae* with each of four SCX fractions of *E. coli* fractions. Error bars indicate the standard error.

adjustments, score cut-offs, or other quality metrics may complicate the interpretation of the test results.

## How to restore the decoy—false positive relationship

The initial idea that proteins which are observable as part of a protein heteromeric crosslink are likely also observable via self crosslinks (Lenz et al, 2021) or monolinks (Parfentev et al, 2020; Zhong et al, 2020) appears sensible. Hence, filtering protein heteromeric matches to proteins that are seen as part of a self crosslink or monolink should reduce noise and hence might improve the detection of protein heteromeric crosslinks. We therefore wondered if the decoy—false positive relationship could be maintained while leveraging this information in post-search filtering.

To accurately estimate the number of false positives after results filtering, it becomes necessary to include an additional set of 'acceptable' proteins. We preserve the relationship between decoy and false matches by considering the decoy complement for each

protein identified with self crosslinks or monolinks (Fig. 2A). This means that for every target protein that passes the initial filter, any crosslink involving the corresponding decoy protein derived from that target protein is also accepted. For example, if protein A is identified, the reversed form of protein A is accepted as passing the filter as well. Similarly, for decoy proteins, we accept the original target protein. This approach ensures a balanced target-decoy relationship for heteromeric proteins even post-filtering. Our modification shows that both tests in Fig. 2B,C are consistent without contradictions. However, there is a noticeable reduction in the total estimated true positives when applying this expected crosslinked proteins filter (ec-filter). This decrease is due to the additional criteria required for identifying crosslinked proteins, which disproportionately affects small and low-abundance proteins due to their lower likelihood of peptide identification (Lenz et al, 2021).

Having established a filter that maintains decoys as a model of both false positive groups, we then evaluated the extent to which this ec-filter improves the number of protein heteromeric matches.

For this, we searched a BS3 crosslinked 26S proteasome dataset with increasing numbers of proteins. First, we used three increasingly larger databases comprising only proteins identified as part of a standard MaxQuant search and then searched against databases additionally supplemented with proteins not previously identified (Fig. 3). On its own, the ec-filter shows possibly a mild improvement when just searching the most abundant proteins. The improvement becomes somewhat more apparent when including more of the lower abundant proteins. However, the ec-filter starts to gain a distinctive advantage when a large extent of non-identified proteins (as a model for non-crosslink-observable) are added to the search database.

The effectiveness of the ec-filter might initially appear counter-intuitive due to the loss of true positives depicted in Fig. 2C. However, the ec-filter is ultimately advantageous. The key difference is that Fig. 3 only considers matches that meet a 5% FDR threshold, whereas Fig. 2C accounts for all estimated true positives. The application of the ec-filter does lead to a reduction in true positives, but it also results in a more significant decrease in false positives. This trade-off contributes to an overall improvement in data quality, which is especially beneficial when analysing many proteins that may not be detectable as part of a crosslink.

As an alternative approach of post-search results optimisation, xiFDR includes a boosting option. This feature increases the number of true positives that pass a specific confidence level by employing a combination of lower-level FDR filters and additional subscores (Fischer and Rappsilber, 2017). When this boosting option is utilised, the advantages of the ec-filter become less pronounced. However, a slight benefit of using the ec-filter may be observed in cases where the database has a substantial surplus of proteins that are either absent or not crosslink-observable in the sample. Thus, when analysing large databases, it might be beneficial to compare results with and without the ec-filter, even when boosting is applied. Both approaches are covered by valid error estimation, allowing users to choose the option that identifies more links at the desired FDR threshold.

## Discussion

Our study emphasises the importance of understanding how filters influence the relationship between decoys and false positives, and the need to adjust for any alterations in this relationship. The recently introduced mi-filter disrupts the balanced relationship between decoys and targets. Similar issues have been noted previously with the target-decoy approach in linear proteomics (Gupta et al, 2011; Debrie et al, 2023), albeit the solutions proposed there do not translate to crosslinks. In contrast, we present the ec-filter, which is based on the same observed patterns that proteins are more likely to be detected in a protein heteromeric crosslink if they have already been identified in self crosslinks or monolinks. This ec-filter effectively increases the detection of protein hetero-meric matches at a given confidence level, without underestimating the incidence of false positives.

The relevance of our findings goes beyond the realm of post-search filtering. Generally, using information about individual proteins for the assessment of protein pairs, both derived from within the search and derived from external sources, has to be done with great care. Search engines or post-processing tools such as ECL-PF (Zhou et al, 2023) and CRIMP 2.0 (Crowder et al, 2023) utilise protein self crosslinks to assign scores or confidence values to protein heteromeric matches. XLinkProphet (Keller et al, 2019) uses information about individual proteins being present to rescore matches. It is not always clear from the available descriptions what is done exactly. However, the developers now have the tools to ensure the correct handling of information: When a protein is assigned a higher confidence level, the same treatment must be applied to its corresponding decoy complement, following the principles of the ec-filter.

We would like to highlight the need for caution when incorporating external information into data analysis processes, especially during different stages of the analysis itself. For instance, when using both the residue-pair-level false discovery rate (residue pair FDR) and the protein–protein interaction-level false discovery rate (PPI-FDR), it's essential to complete the residue-pair FDR assessment before moving on to the PPI-FDR. More broadly, when error assessments are performed sequentially across multiple consolidation levels—where the output of one FDR filter serves as the input for the next—it's crucial to follow the natural order of these levels, such as from crosslinked spectrum matches (CSMs) to peptide pairs, then to residue pairs, and finally to PPIs. Reversing this order, by addressing PPI-FDR first and then the residue pair FDR, risks repeating the flaws seen in the mi-filter approach. This method only considers a subset of remaining errors and can undermine the overall accuracy of the analysis.

We have implemented our current understanding of proper decoy-based error management in crosslinking, including the ec-filter, into the open-source, error-estimation software xiFDR, available in version 2.2 (Fig. EV1, https://www.rappsilberlab.org/software/xifdr/). We encourage the community to openly

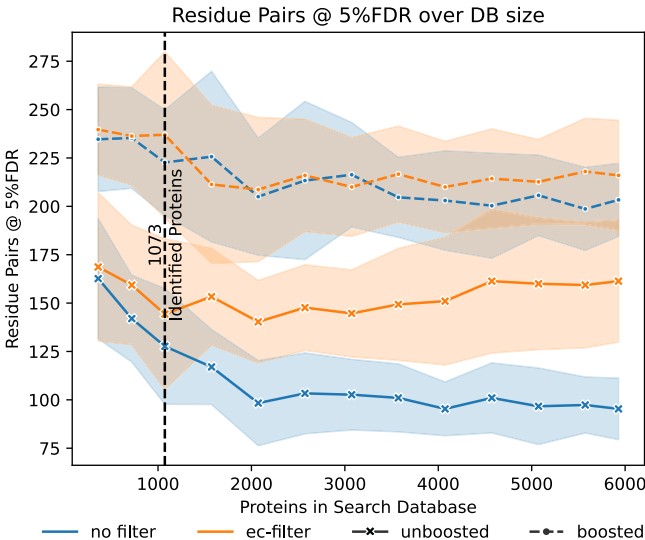

**Figure 3. Effects of ec-filter in xiFDR.**

Results of the xiFDR 2.2 implementation of the ec-filter at a fixed 5%-residue-pair FDR, depending on the initially searched database size. Data plotted are the average results among three fractions searched individually against the same database. The coloured area represents the standard error. The data were searched first against 360, 719, and 1073 present proteins and then against 1073, plus an increasing number of non-present proteins.

communicate any updates or enhancements to FDR estimation to the xiFDR GitHub repository (https://github.com/Rappsilber-Laboratory/xiFDR). The principles of Findability, Accessibility, Interoperability, and Reusability (FAIR) apply specifically to data access. However, the actual use of data critically relies on trust, which in turn depends on robust error management. Achieving this level of trust and accuracy is a collective endeavour that requires the active participation of the entire crosslinking mass spectrometry community.

## Data availability

The datasets and computer code produced in this study are available in the following databases: *Mycoplasma pneumoniae* raw data: ProteomeXchange/Pride ID PXD017711 (https://www.ebi.ac.uk/pride/archive/projects/PXD017711). *Escherichia coli* raw data: ProteomeXchange/JPOST ID JPST000845. 26S proteasome raw data: ProteomeXchange/Pride ID PXD008550 (https://www.ebi.ac.uk/pride/archive/projects/PXD008550). Source code xiSEARCH software: GitHub (https://github.com/Rappsilber-Laboratory/xisearch). Source code xiFDR software: GitHub (https://github.com/Rappsilber-Laboratory/xiFDR). All newly regenerated search/fdr results: zenodo ID 10887761 (https://doi.org/10.5281/zenodo.10887761, https://zenodo.org/records/10887761).

The source data of this paper are collected in the following database record: biostudies:S-SCDT-10_1038-S44320-024-00057-2.

## Peer review information

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

## Acknowledgements

We would like to gratefully acknowledge Dr. Colin Combe and Dr. Andrea Graziadei for fruitful discussions regarding the manuscript and FDR in crosslinking. This research was supported by the Deutsche Forschungsgemeinschaft (DFG, German Research Foundation) under Germany´s Excellence Strategy—EXC 2008—390540038—UniSysCat. The Wellcome Centre for Cell Biology is supported by core funding from the Wellcome Trust [203149]. Open Access funding enabled and organised by Projekt DEAL.

## Author contributions

**Lutz Fischer**: Conceptualisation; Resources; Data curation; Software; Formal analysis; Validation; Investigation; Visualisation; Methodology; Writing—original draft; Project administration; Writing—review and editing. **Juri Rappsilber**: Conceptualisation; Supervision; Funding acquisition; Investigation; Visualisation; Methodology; Writing—original draft; Project administration; Writing—review and editing.

Source data underlying figure panels in this paper may have individual authorship assigned. Where available, figure panel/source data authorship is listed in the following database record: biostudies:S-SCDT-10_1038-S44320-024-00057-2.

## Funding

## Disclosure and competing interests statement

The authors declare no competing interests.

# Expanded View Figure

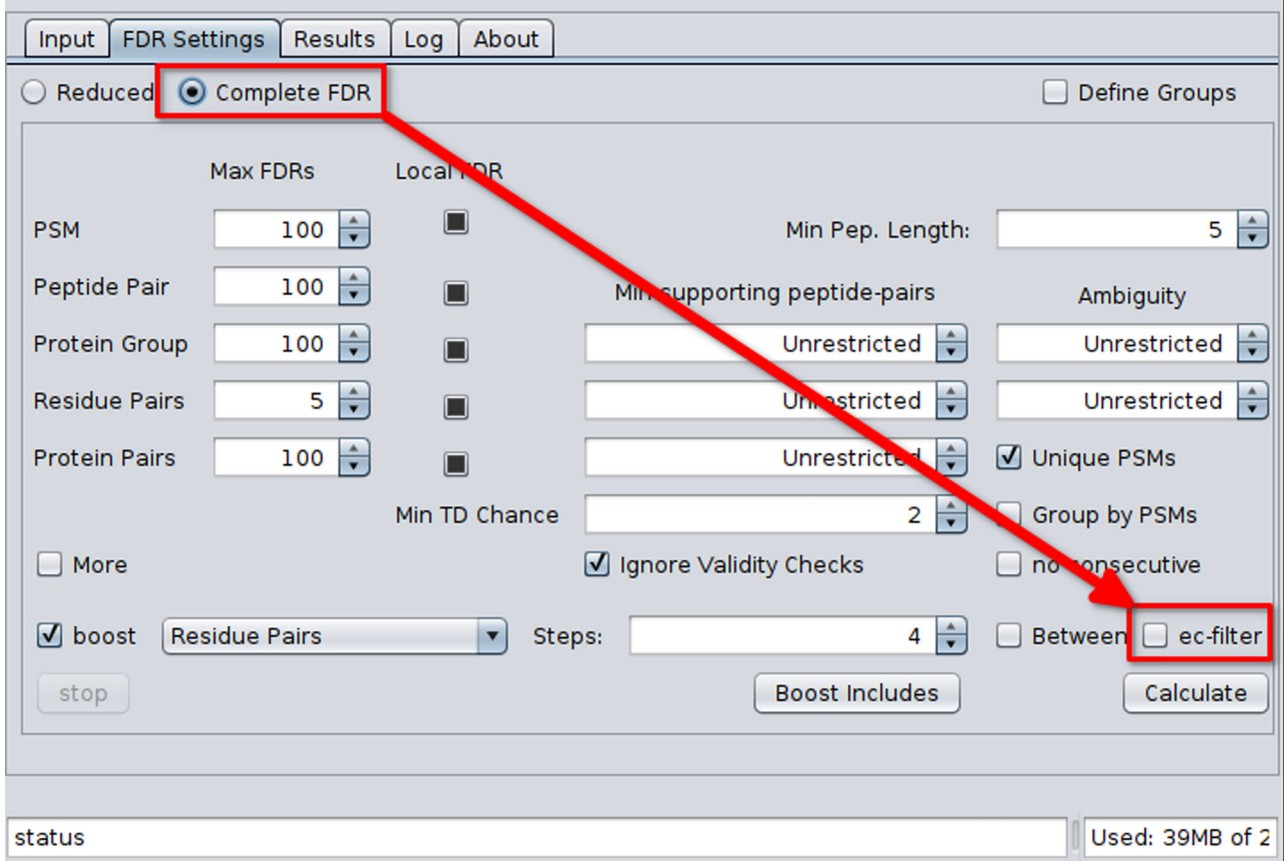

**Figure EV1.  xiFDR ec-filter selection.**

To use the ec-filter the complete settings need to be used and the ec-filter checkbox ticked.

