## [Peer Review File · Molecular Systems Biology]

Rescuing error control in crosslinking mass spectrometry

Juri Rappsilber and Lutz Fischer

Corresponding author(s): Juri Rappsilber (Juri.Rappsilber@TU-Berlin.de)

Review Timeline:

Submission Date:	17th Dec 23
Editorial Decision:	1st Feb 24
Revision Received:	6th May 24
Editorial Decision:	25th Jun 24
Revision Received:	2nd Jul 24
Accepted:	19th Jul 24

Editors: Maria Polychronidou and Jingyi Hou

Transaction Report:

5th Feb 2024

Manuscript Number: MSB-2023-12177

Title: Rescuing error control in crosslinking mass spectrometry

Dear Juri,

Thank you again for submitting your work to Molecular Systems Biology. We have now heard back from the three reviewers who agreed to evaluate your study. Overall, the reviewers acknowledge that the study addresses a relevant problem. Reviewers #1 and #2 are overall more supportive, and mention that while the advance is primarily technical, it is a relevant contribution to the field. As you will see below, reviewer #3 is more critical and thinks that further analyses, with a different design, would be required to perform a fairer comparison of mi-filter and ec-filter. We would invite you to address the concerns of the reviewers in a revision.

I think that the recommendations of the reviewers seem clear and I therefore do not see the need to repeat any of their comments here. During our cross-commenting process (where the reviewers get the chance to make additional comments based on each others' reports), reviewer #2 mentioned the following regarding the issue raised by reviewer #3 on the *E. coli* and *M. pneumoniae* example used to illustrate the false positives: "I think that the example has value, but it clearly also needs to be explained more as it is not very widely used. The authors should then also take the opportunity to find other ways of illustrating the problem without referring to a two-species system. Regarding the relevance in "real-world" application scenarios for crosslinking two proteomes I could point to some studies on host-pathogen interactions that have appeared in the literature." We hope that these additional comments are helpful. In line with some of the comments of the reviewers (e.g. the comment of reviewer #2 recommending introducing the concept of entrapment searches), we would ask you to make sure that the content of the manuscript is easily accessible to the readers. This is particularly important given the technical nature of the study. All issues raised by the reviewers need to be satisfactorily addressed. As you may already know, our editorial policy allows in principle a single round of major revision, so it is essential to provide responses to the reviewers' comments that are as complete as possible. Please feel free to contact me in case you would like to discuss in further detail any of the issues raised or if you would like to share your revision plan with me. I would be happy to schedule a call.

On a more editorial level, we would ask you to address the following points:

- Please include 5 keywords.
- Please provide a .doc version of the manuscript text (including legends for the main figures) and individual production quality figure files for the main Figures (one file per figure).
- We have replaced Supplementary Information by the Expanded View (EV format). In this case, all additional figures can be provided as EV Figures. Please provide one file per EV Figure. Their legends should be included in the manuscript text. For detailed instructions regarding expanded view please refer to our Author Guidelines: .
- In case you include long and complex tables in the Revision, they need to be provided as EV Datasets. Please provide one file per dataset and include the description of each EV Dataset in the dataset file itself, ie. in a separate tab for .xls files or as a README.txt file in .zip folders.
- Please provide a "standfirst text" summarizing the study in one or two sentences (approximately 250 characters), three to four "bullet points" highlighting the main findings and a "synopsis image" (550px width and max 400px height, jpeg format) to highlight the paper on our homepage.
- All Materials and Methods need to be described in the main text. We would ask you to use 'Structured Methods', our new Materials and Methods format, which is mandatory for Methods and Articles with a strong methodological focus. According to this format, the Materials and Methods section should include a Reagents and Tools Table (listing key reagents, experimental models, software and relevant equipment and including their sources and relevant identifiers) followed by a Methods and Protocols section in which we encourage the authors to describe their methods using a step-by-step protocol format with bullet points, to facilitate the adoption of the methodologies across labs. More information on how to adhere to this format as well as downloadable templates (.doc or .xls) for the Reagents and Tools Table can be found in our author guidelines: . An example of a Method paper with Structured Methods can be found here: .
- Please include a "Disclosure and Competing Interests Statement" in the main text.
- Please include a Data availability section describing how the data, code etc. have been made available. This section needs to be formatted according to the example below:

The datasets and computer code produced in this study are available in the following databases:

- Chip-Seq data: Gene Expression Omnibus GSE46748 (<https://www.ncbi.nlm.nih.gov/geo/query/acc.cgi?acc=GSE46748>)
- Modeling computer scripts: GitHub (<https://github.com/SysBioChalmers/GECKO/releases/tag/v1.0>)
- [data type]: [full name of the resource] [accession number/identifier] ([doi or URL or identifiers.org/DATABASE:ACCESSION])

- The References should be formatted according to the Molecular Systems Biology reference style (i.e., ordered alphabetically and listing the first 10 authors followed by et al).

- Molecular Systems Biology supports formal data citations in the Reference list, to cite previously published datasets. In addition to citing the original papers that reported the data, we encourage you to also cite the relevant datasets directly in the Reference list. In the text, references to datasets are included as "Data ref: Smith et al, 2001" or "Data ref: NCBI Sequence Read Archive PRJNA342805, 2017". In the Reference list, data citations are very similar to normal literature references but must be labeled with "[DATASET]" at the end of the reference. For detailed instructions please refer to our Author Guidelines .

- For data quantification: please specify the name of the statistical test used to generate error bars and P values, the number (n) of independent experiments (specify technical or biological replicates) underlying each data point and the test used to calculate p-values in each figure legend. The figure legends should contain a basic description of n, P and the test applied. Graphs must include a description of the bars and the error bars (s.d., s.e.m.).

- When you resubmit your manuscript, please download our CHECKLIST (<https://bit.ly/EMBOPressAuthorChecklist>) and include the completed form in your submission.

Please note that the Author Checklist will be published alongside the paper as part of the transparent process (<https://www.embopress.org/page/journal/17444292/authorguide#transparentprocess>).

If you feel you can satisfactorily deal with these points and those listed by the referees, you may wish to submit a revised version of your manuscript. Please attach a covering letter giving details of the way in which you have handled each of the points raised by the referees. A revised manuscript will be once again subject to review and you probably understand that we can give you no guarantee at this stage that the eventual outcome will be favorable.

Kind regards,

Maria

Maria Polychronidou, PhD
Senior Editor
Molecular Systems Biology

We realize that it is difficult to revise to a specific deadline. In the interest of protecting the conceptual advance provided by the work, we recommend a revision within 3 months (1st May 2024). Please discuss the revision progress ahead of this time with the editor if you require more time to complete the revisions. Use the link below to submit your revision:

IMPORTANT:

See also figure legend guidelines: <https://www.embopress.org/page/journal/17444292/authorguide#figureformat>

- Please note that corresponding authors are required to supply an ORCID ID for their name upon submission of a revised manuscript (EMBO Press signed a joint statement to encourage ORCID adoption).

(<https://www.embopress.org/page/journal/17444292/authorguide#editorialprocess>)

Currently, our records indicate that the ORCID for your account is 0000-0001-5999-1310.

Link Not Available

*** PLEASE NOTE *** As part of the EMBO Press transparent editorial process initiative (see our Editorial at <https://dx.doi.org/10.1038/msb.2010.72>), Molecular Systems Biology publishes online a Review Process File with each accepted manuscripts. This file will be published in conjunction with your paper and will include the anonymous referee reports, your point-by-point response and all pertinent correspondence relating to the manuscript. If you do NOT want this File to be published, please inform the editorial office at msb@embo.org within 14 days upon receipt of the present letter.

Reviewer #1:

Summary

The paper titled "Rescuing error control in crosslinking mass spectrometry" focuses on improving the accuracy and reliability of protein-protein interaction (PPI) data in crosslinking mass spectrometry experiments.

The title of the manuscript sounds very promising. However, I would also like to point out that the expectations of such an excellent laboratory as the Rappasilber laboratory are very high. I have therefore evaluated this manuscript very critically.

The authors highlight the issues with the current heuristic approaches used to improve identification rates in crosslinking MS, in this case the mi-filter. These approaches can undermine error estimation and lead to a significant underestimation of false positives, thereby affecting the reliability of PPI identification. The study utilized freely available datasets from ProteomeXchange and examined crosslinking MS data from different sources, including *E. coli* and *M. pneumoniae* crosslinked with DSSO, to assess the accuracy of error estimates affected by post-search-filters. The authors describe two possible groups of false positives to base their assumptions and new filter on. Group 1 comprises random matches of at least one protein that is not observable as part of a crosslink and group 2 observable crosslinks between two proteins that are correct in combination but can be wrong in other combinations.

To overcome issues with the current mi-filter in error estimation, the authors proposed a new approach, the ec-filter, which maintains a balanced relationship between decoy and false matches. This method allows for a more accurate estimation of false positives, ensuring that the identifications of crosslinks and especially the heteromeric ones are more reliable. Additionally, the authors evaluated the extent to which the ec-filter improves the number of protein heteromeric matches. Here they used a different publicly available dataset of the 26S proteasome crosslinked with BS3. The crosslink search was performed with increasing database sizes.

First databases comprised only identified proteins by MaxQuant, followed by databases that included an increasing amount of not identified proteins. The effect of the ec-filter increased with large databases including a high number of not identified proteins but was minimal when using the boosting function of xiFDR.

General Remarks

The study appears convincing due to its methodical approach to addressing a significant problem in crosslinking mass spectrometry, absolutely. By empirically testing and demonstrating the pitfalls of existing methods and proposing a solution, the study offers valuable insights. This work is significant in the context of protein-protein interaction studies, where accurate identification of interactions is crucial to evaluate biological findings and assemblies of protein complexes.

The advance here is primarily technical, enhancing the accuracy of error estimations in crosslinking experiments. This advance is significant as it addresses a fundamental challenge in crosslinking mass spectrometry, potentially impacting the reliability of many studies in this field. The audience for this study includes researchers in proteomics, bioinformatics, and related areas of molecular biology. It specifically addresses researchers of crosslinking community as well as a lot of proteomics core facilities to beware of correct error handling to produce and share reliable crosslinking data.

The false positive rate is a problem that is still very much underestimated. If the false positive rate is misjudged, this results in a very high number of protein-protein - which is nice to show for a publication. However, these values are not usable in practice - especially in core facilities - where a misinterpreted cross-link can cause a lot of work in biology. Anywhere, my group has already published a number of publications on this important subject.

To summarize, I would like to say that my high expectations were absolutely fulfilled with this manuscript, and I would suggest a minor revision.

Minor Points

The paper is well-structured with clear sections. However, there are minor mistakes and remarks:

Method section:

- It would be nice to describe what kind of settings have been used to convert raw files to mgf files using MSConvert or cite a paper where it is has been described for crosslinking data. The paper cited here is just to cite MS Convert.
- please state what kind of crosslinking search engine was used. It is clear to me that presumably xiSEARCH was used but it would be interesting to know which version. Same for xiFDR (version is just mentioned in the legend of figure 4)
- please also state the bias towards Lysine and n-terminal, usually you can apply a penalty for STY in xiSEARCH, please state the correct number used or state what "bias towards lysine and n-term" means here
- please add the size of the database sets that were used to assess the effect of ec-filter, it's not clear to me what the smallest data set is if looking at figure 4 only

Figure 1 and 3:

- please translate "Decoy Protein basierent auf X" into English in the legend

Figure 2:

- please state something about the filter loss for the ec-filter, either in the discussion or results section. It seems that in Figure 4, even for the smallest database with no additional non-identified proteins, the filter loss disappears. Why is that the case? Is it because of a different data set? Or differences between cleavable and non-cleavable crosslinker?

Discussion section:

- please add to the discussion your opinion on when the ec-filter should be used. As I understood it, if I use the boosting function in xiFDR the effect of the ec-filter is small even for high amounts of non-identified proteins in the database. So, when does it make sense to use the ec-filter instead of the boosting function for PPI-crosslinking projects?

Many thanks to all authors for keeping track of error estimation strategies in crosslinking mass spectrometry. This publication is very helpful for the scientific community and will certainly be cited very often.

Reviewer #2:

Fischer and Rappsilber discuss a recent publication from Stengel and co-workers (Chen et al.) on the use of post-search filtering criteria to reduce the search space in crosslinking mass spectrometry data analysis. They demonstrate that the "mi-filter" proposed by Chen et al. violates central principles of the target/decoy search strategy and leads to a sometimes severe underestimation of FDRs. The authors use this example as a starting point to address the search space problem and different strategies for FDR control.

Major comments:

In this manuscript, the authors address valid points about target-decoy competition in data analysis. However, it would be important to mention that this problem is neither new nor restricted to crosslinking data. An early example is the second-pass search of X!Tandem, which leads to a similar problem as the mi-filter. This has been discussed in many works about a decade ago, including this commentary by Gupta et al. (DOI:10.1007/s13361-011-0139-3), and was recently taken up again by Debrie et al. (DOI:10.1021/acs.jproteome.2c00423). The essential requirement for any post-search filter criteria is that they be applied to target and decoy populations alike. The mi-filter approach violates this principle, as Fischer and Rappsilber confirm.

The manuscript mainly addresses "deficiencies" in a publication from Analytical Chemistry, although it then discusses the problem in a somewhat broader context. However, why not publish a comment in that journal, which would probably require a response from the original authors?

In the Methods section, the authors never mention clearly that xiSEARCH was used for all searches (I assume), and which version of the software was used. The concept of entrapment searches could also be stated more clearly in this section; it is only discussed on page 4, and not all readers might be familiar with this concept.

Overall, Figure 2 is hard to understand. It maybe would be helpful to also give numbers and not only percentages in A and C. In A, what would be the corresponding numbers/percentages if FDR control is performed without the mi-filter?

In Figure 4, the authors show that an adapted and compliant filter performs comparable to xiFDRs "boosting" strategy. However, the Rappsilber group also frequently employs search space restrictions based on protein quantities (iBAQ) in their works. How would this approach compare here?

Minor comments:

p. 1: Looking at reference 3, I could not confirm that the heteromer and self-link terminology is discussed there.

p. 1: What do the authors mean by "noise levels" in this context?

p. 4: Why would the mi-filter lead to the "exclusion of small proteins"? I assume it would rather be biased against proteins with few identifiable peptides.

p. 7: The authors should mention that in Ref. 8, the "boosting" approach of xiFDR was not called like that, so it is not immediately obvious that the two approaches are identical (boosting and pre-filter).

p. 9: Ref. 21 is published: <https://pubs.acs.org/doi/full/10.1021/acs.analchem.3c00329>

Figures: Some residual German text in the legends of Figs. 1 and 3 ("basierent auf"); "Missmatch" > "Mismatch" in Figure 2.

Reviewer #3:

Fischer and Rappsilber argue that the recently-described mi-filter (PMID: 36510358) is very significantly underestimating the false positive fraction in inter-protein crosslink detection. Their argument is based on an analysis pipeline (designed specifically to this manuscript) in which the mi-filter is applied on XLMS data that are a combination of two separate XLMS studies from two bacteria. They then suggest a variation of the mi-filter (termed ec-filter) that corrects for the alleged underestimate.

I first want to emphasize that I completely concur with the warnings of the Rappsilber lab of the need for better estimates of the FDRs of inter-protein crosslinks. For example, in my experience pLink is notorious in its poor FDR estimates of such crosslinks. However, I strongly disagree with their argument about the mi-filter as presented in the manuscript. Their devised pipeline (which is not a realistic experimental setup of a biological study) is violating the basic assumption underlining mi-filter. The violation stems from passing *M. pneumoniae* proteins through the mi-filter while analyzing *E. coli* spectra (and vice versa). This would not have occurred in a typical biological study in which only one species was studied. It is therefore not surprising that under this violation the mi-filter "breaks" and many dataset-mismatches are observed. I would guess that have they included results of the mi-filter in a more realistic analysis scenario, such as the one shown in Figure 4, the mi-filter will show a reasonable FDR behavior that I predict will be on par with their ec-filter.

In addition, I found the presentation of results in Fig. 2 to be very hard to understand, and also lacking the dependence on the cut-off score. I would have preferred if they presented their results in graphs similar to Figs. 2 and 3 in the mi-filter paper.

Reviewer #1:**Summary**

The paper titled "Rescuing error control in crosslinking mass spectrometry" focuses on improving the accuracy and reliability of protein-protein interaction (PPI) data in crosslinking mass spectrometry experiments.

The title of the manuscript sounds very promising. However, I would also like to point out that the expectations of such an excellent laboratory as the Rappsilber laboratory are very high. I have therefore evaluated this manuscript very critically.

The authors highlight the issues with the current heuristic approaches used to improve identification rates in crosslinking MS, in this case the mi-filter. These approaches can undermine error estimation and lead to a significant underestimation of false positives, thereby affecting the reliability of PPI identification. The study utilized freely available datasets from ProteomeXchange and examined crosslinking MS data from different sources, including *E. coli* and *M. pneumoniae* crosslinked with DSSO, to assess the accuracy of error estimates affected by post-search-filters. The authors describe two possible groups of false positives to base their assumptions and new filter on. Group 1 comprises random matches of at least one protein that is not observable as part of a crosslink and group 2 observable crosslinks between two proteins that are correct in combination but can be wrong in other combinations. To overcome issues with the current mi-filter in error estimation, the authors proposed a new approach, the ec-filter, which maintains a balanced relationship between decoy and false matches. This method allows for a more accurate estimation of false positives, ensuring that the identifications of crosslinks and especially the heteromeric ones are more reliable. Additionally, the authors evaluated the extent to which the ec-filter improves the number of protein heteromeric matches. Here they used a different publicly available dataset of the 26S proteasome crosslinked with BS3. The crosslink search was performed with increasing database sizes.

First databases comprised only identified proteins by MaxQuant, followed by databases that included an increasing amount of not identified proteins. The effect of the ec-filter increased with large databases including a high number of not identified proteins but was minimal when using the boosting function of xiFDR.

General Remarks

The study appears convincing due to its methodical approach to addressing a significant problem in crosslinking mass spectrometry, absolutely. By empirically testing and demonstrating the pitfalls of existing methods and proposing a solution, the study offers valuable insights. This work is significant in the context of protein-protein interaction studies, where accurate identification of interactions is crucial to evaluate biological findings and assemblies of protein complexes.

The advance here is primarily technical, enhancing the accuracy of error estimations in crosslinking experiments. This advance is significant as it addresses a fundamental challenge in crosslinking mass spectrometry, potentially impacting the reliability of many studies in this field. The audience for this study includes researchers in proteomics, bioinformatics, and related areas of molecular biology. It specifically addresses researchers of crosslinking community as well as a lot of proteomics core facilities to beware of correct error handling to produce and share reliable crosslinking data.

The false positive rate is a problem that is still very much underestimated. If the false positive rate is misjudged, this results in a very high number of protein-protein - which is nice to show for a publication. However, these values are not usable in practice - especially in core facilities - where a misinterpreted cross-link can cause a lot of work in biology. Anywhere, my group has already published a number of publications on this important subject.

To summarize, I would like to say that my high expectations were absolutely fulfilled with this manuscript, and I would suggest a minor revision.

Minor Points

The paper is well-structured with clear sections. However, there are minor mistakes and remarks:

Method section:

- It would be nice to describe what kind of settings have been used to convert raw files to mgf files using MSConvert or cite a paper where it has been described for crosslinking data. The paper cited here is just to cite MS Convert.

We changed the text to: The raw files were converted to mgf-files with msConvert from ProteoWizard (version 3.0) (Chambers *et al*, 2012) **with peak picking enabled.**

- please state what kind of crosslinking search engine was used. It is clear to me that presumably xiSEARCH was used but it would be interesting to know which version. Same for xiFDR (version is just mentioned in the legend of figure 4)

We added the version of xiSEARCH (**Version 1.7.6.4**) and xiFDR (2.2) to the text.

- please also state the bias towards Lysine and n-terminal, usually you can apply a penalty for STY in xiSEARCH, please state the correct number used or state what "bias towards lysine and n-term" means here

We changed the description to: **"crosslinker DSSO for M. pneumoniae and E. coli dataset and BS3 for the 26S proteasome dataset with specificity for Lysine, Serine, Threonine, Tyrosine and protein n-terminal with a penalty value for Serine, Threonine, and Tyrosine of 0.2"**

- please add the size of the database sets that were used to assess the effect of ec-filter, it's not clear to me what the smallest data set is if looking at figure 4 only

We added "The data were searched first against 360, 719 and 1073 present proteins and then against the 1073, plus an increasing number of non-present proteins." to the figure legend.

Figure 1 and 3:

- please translate "Decoy Protein basierend auf X" into English in the legend
This has been corrected.

Figure 2:

- please state something about the filter loss for the ec-filter, either in the discussion or results section.

We added **"However, there is a noticeable reduction in the total estimated true positives when applying this expected crosslinked proteins filter (ec-filter). This decrease is due to the additional criteria required for identifying crosslinked proteins, which disproportionately affects small and low-abundance proteins due to their lower likelihood of peptide identification (Lenz et al, 2021)."**

It seems that in Figure 4, even for the smallest database with no additional non-identified proteins, the filter loss disappears. Why is that the case? Is it because of a different data set? Or differences between cleavable and non-cleavable crosslinker?

Figure 2C plots the estimated TPs among all matches (but makes no statement on how many we can actually separate from noise), while Figure 3 (former Figure 4) plots what passes 5% FDR. Applying the ec-filter reduces the estimated TPs among all matches (seen in Fig 2C) but increases the separation between TPs and noise. In Figure 3, one can see that the losses and gains essentially balance for the smallest database.

We changed the associated text to: **"The effectiveness of the ec-filter might initially appear counterintuitive due to the loss of true positives depicted in Figure 2C. However, the ec-filter is ultimately advantageous. The key difference is that Figure 3 only considers matches that meet a 5% FDR threshold, whereas Figure 2C accounts for all estimated true positives. The application of the ec-filter does lead to a reduction in true positives, but it also results in a more significant decrease in false positives. This trade-off contributes to an overall improvement in data quality, which is especially beneficial when analysing many proteins that may not be detectable as part of a crosslink."**

Discussion section:

- please add to the discussion your opinion on when the ec-filter should be used. As I understood it, if I use the boosting function in xiFDR the effect of the ec-filter is small even for high amounts of non-identified proteins in the database. So, when does it make sense to use the ec-filter instead of the boosting function for PPI-crosslinking projects?

We added: **“Thus, when analysing large databases, it might be beneficial to compare results with and without the ec-filter, even when boosting is applied. Both approaches are covered by valid error estimation, allowing users to choose the option that identifies more links at the desired FDR threshold.”**

Many thanks to all authors for keeping track of error estimation strategies in crosslinking mass spectrometry. This publication is very helpful for the scientific community and will certainly be cited very often.

We would like to thank the reviewer for their supportive words and valuable suggestions to improve our manuscript.

Reviewer #2:

Fischer and Rappsilber discuss a recent publication from Stengel and co-workers (Chen et al.) on the use of post-search filtering criteria to reduce the search space in crosslinking mass spectrometry data analysis. They demonstrate that the "mi-filter" proposed by Chen et al. violates central principles of the target/decoy search strategy and leads to a sometimes severe underestimation of FDRs. The authors use this example as a starting point to address the search space problem and different strategies for FDR control.

Major comments:

In this manuscript, the authors address valid points about target-decoy competition in data analysis. However, it would be important to mention that this problem is neither new nor restricted to crosslinking data. An early example is the second-pass search of X!Tandem, which leads to a similar problem as the mi-filter. This has been discussed in many works about a decade ago, including this commentary by Gupta et al. (DOI:10.1007/s13361-011-0139-3), and was recently taken up again by Debrie et al. (DOI:10.1021/acs.jproteome.2c00423). The essential requirement for any post-search filter criteria is that they be applied to target and decoy populations alike. The mi-filter approach violates this principle, as Fischer and Rappsilber confirm.

The authors of the mi-filter would probably argue that they adhere to “The essential requirement for any post-search filter criteria is that they be applied to target and decoy populations alike.” It is therefore important to establish that their procedure is breaking the target-decoy relationship. We would like to raise the reviewer’s attention to the fact that we provide tests and solutions in addition to pointing at the problem. This can also be applied to linear proteomics. However, the effects are particularly consequential and felt in crosslinking. We thank the reviewer for bringing to our attention the work of Gupta et al which points out cases in normal proteomics where one does treat target and decoys on first sight equal but perturbs the target-decoy equivalence by taking match-external information into account. However, their solution does not work with the current scoring methods used in crosslinking MS. Nevertheless, we have added this reference now to our manuscript.

The manuscript mainly addresses "deficiencies" in a publication from Analytical Chemistry, although it then discusses the problem in a somewhat broader context. However, why not publish a comment in that journal, which would probably require a response from the original authors?

We aim at a fundamental problem and provide a test and solution to it. While this problem can be found in the mi-filter, it exists very likely in other commonly used software packages as well. The information of proteins being “crosslink observable” is used e.g. in XLinkProphet, ECL-PF and others. Our work raises attention to the problem and empowers developers and users alike to test their current and also future tools through a relatively simple control.

In the Methods section, the authors never mention clearly that xiSEARCH was used for all searches (I assume), and which version of the software was used. The concept of entrapment searches could also be stated more clearly in this section; it is only discussed on page 4, and not all readers might be familiar with this concept.

We added the version information and added the following sentence: “**This methodology is akin to a traditional entrapment strategy, where a dataset is searched against both target and non-present (entrapment) protein sequences. The advantage of the pairwise entrapment model is that all proteins act simultaneously as targets for some spectra and as known false protein for others.**”

Overall, Figure 2 is hard to understand. It maybe would be helpful to also give numbers and not only percentages in A and C.

These panels display confidence, which is expressed in percent. We now added the count for targets to the figure legends.

In A, what would be the corresponding numbers/percentages if FDR control is performed without the mi-filter?

We added 2 % and 1 % FDR for the non-filtered data.

In Figure 4, the authors show that an adapted and compliant filter performs comparable to xiFDRs "boosting" strategy. However, the Rappsilber group also frequently employs search space restrictions based on protein quantities (iBAQ) in their works. How would this approach compare here?

Any strategy of constructing the search database is a pre-search approach and is complementary to post-search approaches.

Minor comments:

p. 1: Looking at reference 3, I could not confirm that the heteromer and self-link terminology is discussed there. The reference was removed.

p. 1: What do the authors mean by "noise levels" in this context?

We inserted an explanation: “noise levels—the proportion of random matches in these groups”.

p. 4: Why would the mi-filter lead to the "exclusion of small proteins"? I assume it would rather be biased against proteins with few identifiable peptides.

We changed the text to: “**Note that the mi-filter also reduces correct matches by biasing against proteins with few identifiable peptides, for example small proteins (Lenz et al, 2021)**”

p. 7: The authors should mention that in Ref. 8, the "boosting" approach of xiFDR was not called like that, so it is not immediately obvious that the two approaches are identical (boosting and pre-filter).

We have now rephrased this: “As an alternative approach of post-search results optimisation, xiFDR includes a boosting option. This feature increases the number of true positives that pass a specific confidence level by employing a combination of lower-level FDR filters and additional subscores (Fischer & Rappsilber, 2017).“

p. 9: Ref. 21 is published: <https://pubs.acs.org/doi/full/10.1021/acs.analchem.3c00329>

We updated the reference.

Figures: Some residual German text in the legends of Figs. 1 and 3 ("basierent auf"); "Missmatch" > "Mismatch" in Figure 2.

We have corrected them.

Reviewer #3:

Fischer and Rappsilber argue that the recently-described mi-filter (PMID: 36510358) is very significantly underestimating the false positive fraction in inter-protein crosslink detection. Their argument is based on an analysis pipeline (designed specifically to this manuscript) in which the mi-filter is applied on XLMS data that are a combination of two separate XLMS studies from two bacteria. They then suggest a variation of the mi-filter (termed ec-filter) that corrects for the alleged underestimate.

I first want to emphasize that I completely concur with the warnings of the Rappsilber lab of the need for better estimates of the FDRs of inter-protein crosslinks. For example, in my experience pLink is notorious in its poor FDR estimates of such crosslinks. However, I strongly disagree with their argument about the mi-filter as presented in the manuscript. Their devised pipeline (which is not a realistic experimental setup of a biological study) is violating the basic assumption underlining mi-filter. The violation stems from passing *M. pneumoniae* proteins through the mi-filter while analyzing *E. coli* spectra (and vice versa).

The basic assumption underlying the mi-filter is that proteins detected as crosslinked with other proteins also are detected with self links and monolinks. We agree in this basic assumption (although it holds caveats for difficult to observe proteins). However, the implementation is critical. We constructed our test case to make the underlying error in the mi-filter implementation and similar approaches visible that break the equivalence of decoys and false positives. Mixing two species is just a practicable way of reproducing two sets of proteins that we are confident about not interacting (= ground truth is known). Also in a single species there will be such non-interacting sets of proteins, just that we lack ground truth regarding their identity. Imagine analysing a eukaryotic cell (distinct sets of proteins are found in compartments) or a single cell pathogen invading a host cell. Even in a simpler system, there will be distinct protein groups (different protein complexes or proteins within a complex but spatially separate) just that we typically lack ground truth regarding them not interacting.

This would not have occurred in a typical biological study in which only one species was studied. It is therefore not surprising that under this violation the mi-filter "breaks" and many dataset-mismatches are observed. I would guess that have they included results of the mi-filter in a more realistic analysis scenario, such as the one shown in Figure 4, the mi-filter will show a reasonable FDR behavior that I predict will be on par with their ec-filter.

In fact, the same does happen in a single species investigation. It just is not visible because we lack ground truth.

Looking at figure 4, the original mi-filter will probably not just be “on par with ec-filter”, it will *appear* to outperform ec-filter in number of identified target matches, likely by a large margin. This is a consequence of the remaining decoys not representing all false positives. Consequently, any decoy-based FDR that follows mi-filter and similar approaches has to underestimate the false positives among the remaining target proteins.

In addition, I found the presentation of results in Fig. 2 to be very hard to understand, and also lacking the dependence on the cut-off score. I would have preferred if they presented their results in graphs similar to Figs. 2 and 3 in the mi-filter paper.

The point of Figure 2 is to demonstrate that the mi-filter implementation of the underlying idea is flawed. We rearranged the figure to enhance clarity and fused it with the schematics explaining the respective filter approaches.

For the interest of the reviewer, we made a plot reminiscent to the mi-filter paper (Rebuttal Figure 1) - showing that independent of the score cut-off (minscore), the observable error following mi-filter use (mixed TT) is noticeably larger than indicated by the remaining decoys (2 % decoy-derived FDR for all score cut-offs):

Rebuttal Figure 1: (a) Plotted as a function of score cut-off (minscore) is the number of target-target matches (TT) after a 2 % decoy-derived FDR cut-off, and the number of known false matches among the target-target matches, based on our paired entrapment control (mixed TT). (b) Same data as in panel a, but showing percentage.

25th Jun 2024

Manuscript Number: MSB-2023-12177R

Title: Rescuing error control in crosslinking mass spectrometry

Author: Juri Rappsilber

Lutz Fischer

Thank you for sending us your revised manuscript. We have now heard back from the two referees who agreed to evaluate your study. From the comments below, you will see that Referees #1 and #2 are satisfied with the revisions and support the publication of the manuscript. Since we were unable to obtain a report from Referee #3, we sought advice from one of the other referees, who found your response to Referee #3's concerns to be convincing.

Before we can formally accept your manuscript, we would ask you to address the following editorial-level issues:

1. Please include the callouts for Figure 2A, 2B.
2. Please include the manuscript ID number in the author checklist.
3. I have slightly modified the synopsis text (see attached). Please let me know if it is fine as is or if you would like to introduce further modifications.

Please resubmit the paper ****within one month**** and ideally as soon as possible. Please use the Manuscript Number (above) in all correspondence.

When you resubmit your manuscript, please download our CHECKLIST (<https://bit.ly/EMBOPressAuthorChecklist>) and include the completed form in your submission. ***Please note*** that the Author Checklist will be published alongside the paper as part of the transparent process (<https://www.embopress.org/page/journal/17444292/authorguide#transparentprocess>)

Click on the link below to submit your revised paper.

Kind regards,
Jingyi

Jingyi Hou, PhD
Scientific Editor
Molecular Systems Biology

If you do choose to resubmit, please click on the link below to submit the revision online before 25th Jul 2024.

IMPORTANT: When you send your revision, we will require the following items:

1. the manuscript text in LaTeX, RTF or MS Word format
2. a letter with a detailed description of the changes made in response to the referees. Please specify clearly the exact places in the text (pages and paragraphs) where each change has been made in response to each specific comment given
3. three to four 'bullet points' highlighting the main findings of your study

4. a short 'blurb' text summarizing in two sentences the study (max. 250 characters)
5. a 'thumbnail image' (550px width and max 400px height, Illustrator, PowerPoint or jpeg format), which can be used as 'visual title' for the synopsis section of your paper.
6. Please include an author contributions statement after the Acknowledgements section (see <https://www.embopress.org/page/journal/17444292/authorguide#manuscriptpreparation>)
7. Please complete the CHECKLIST available at (<https://bit.ly/EMBOPressAuthorChecklist>). Please note that the Author Checklist will be published alongside the paper as part of the transparent process (<https://www.embopress.org/page/journal/17444292/authorguide#transparentprocess>).
8. When assembling figures, please refer to our figure preparation guideline in order to ensure proper formatting and readability in print as well as on screen:
<https://bit.ly/EMBOPressFigurePreparationGuideline>
See also figure legend guidelines: <https://www.embopress.org/page/journal/17444292/authorguide#figureformat>
9. Please note that corresponding authors are required to supply an ORCID ID for their name upon submission of a revised manuscript (EMBO Press signed a joint statement to encourage ORCID adoption). (<https://www.embopress.org/page/journal/17444292/authorguide#editorialprocess>)
Currently, our records indicate that the ORCID for your account is 0000-0001-5999-1310.

Please click the link below to modify this ORCID:
Link Not Available

*** PLEASE NOTE *** As part of the EMBO Press transparent editorial process initiative (see our Editorial at <https://dx.doi.org/10.1038/msb.2010.72> , Molecular Systems Biology will publish online a Review Process File to accompany accepted manuscripts. When preparing your letter of response, please be aware that in the event of acceptance, your cover letter/point-by-point document will be included as part of this File, which will be available to the scientific community. More information about this initiative is available in our Instructions to Authors. If you have any questions about this initiative, please contact the editorial office (msb@embo.org).

Reviewer #1:

The authors answered all my questions to my complete satisfaction.

Reviewer #2:

In this revised version, the authors have addressed all my comments appropriately. The redesigned figures are easier to follow.

All editorial and formatting issues were resolved by the authors.

19th Jul 2024

Manuscript number: MSB-2023-12177RR

Title: Rescuing error control in crosslinking mass spectrometry

Dear Dr. Rappsilber,

Thank you again for sending us your revised manuscript. We are now satisfied with the modifications made and I am pleased to inform you that your paper has been accepted for publication.

Yours sincerely,

Sincerely,

Jingyi Hou, PhD
Scientific Editor
Molecular Systems Biology
